# Influence of Multiple Thermomechanical Processing of 3D Filaments Based on Polylactic Acid and Polyhydroxybutyrate on Their Rheological and Utility Properties

**DOI:** 10.3390/polym14101947

**Published:** 2022-05-11

**Authors:** Roderik Plavec, Vojtech Horváth, Slávka Hlaváčiková, Leona Omaníková, Martina Repiská, Elena Medlenová, Jozef Feranc, Ján Kruželák, Radek Přikryl, Silvestr Figalla, Soňa Kontárová, Andrej Baco, Lucia Danišová, Zuzana Vanovčanová, Pavol Alexy

**Affiliations:** 1Institute of Natural and Synthetic Polymers, Faculty of Chemical and Food Technology, Slovak University of Technology, Radlinského 9, 812 37 Bratislava, Slovakia; vojtech.horvath@stuba.sk (V.H.); slavka.hlavacikova@stuba.sk (S.H.); leona.omanikova@stuba.sk (L.O.); martina.repiska@stuba.sk (M.R.); elena.medlenova@stuba.sk (E.M.); jozef.feranc@stuba.sk (J.F.); jan.kruzelak@stuba.sk (J.K.); andrej.baco@stuba.sk (A.B.); lucia.danisova@stuba.sk (L.D.); zuzana.vanovcanova@stuba.sk (Z.V.); pavol.alexy@stuba.sk (P.A.); 2Institute of Materials Chemistry, Faculty of Chemistry, Brno University of Technology, Purkyňova 464/118, 612 00 Brno, Czech Republic; prikryl@fch.vut.cz (R.P.); xfigalla@fc.vut.cz (S.F.); kontarova@fch.vut.cz (S.K.)

**Keywords:** material recycling, polyhydroxybutyrate, polylactic acid, 3D printing

## Abstract

This study focused on material recycling of a biodegradable blend based on PLA and PHB for multiple applications of biodegradable polymeric material under real conditions. In this study, we investigated the effect of multiple processing of a biodegradable polymer blend under the trade name NONOILEN^®^, which was processed under laboratory as well as industrial conditions. In this article, we report on testing the effect of blending and multiple processing on thermomechanical stability, molecular characteristics, as well as thermophysical and mechanical properties of experimental- and industrial-type tested material suitable for FDM 3D technology. The results showed that the studied material degraded during blending and subsequently during multiple processing. Even after partial degradation, which was demonstrated by a decrease in average molecular weight and a decrease in complex viscosity in the process of multiple reprocessing, there was no significant change in the material’s thermophysical properties, either in laboratory or industrial conditions. There was also no negative impact on the strength characteristics of multiple processed samples. The results of this work show that a biodegradable polymer blend based on PLA and PHB is a suitable candidate for material recycling even in industrial processing conditions. In addition, the results suggest that the biodegradable polymeric material NONOILEN^®^ 3D 3056-2 is suitable for multiple uses in FDM technology.

## 1. Introduction

In recent years, consumer lifestyles have led to a dramatic increase in the consumption of polymers (especially plastics). In 2019, 368 million tons of plastic material were produced worldwide, of which, 57.9 million tons of plastic material were produced in the EU [1].

The use of plastics is continuing to increase which is significantly increasing plastic waste production, and therefore there is greater interest in recycling and reusing plastics. In 2018, 9.4 million tons of plastic waste were recycled in the EU [2,3]. According to the EU directive 2018/852, at least 65% by weight of all packaging waste must be recycled in EU countries by 2025 and up to 70% by weight of all packaging waste by 2030 [4].

In this article, we present the possibility of material recycling of biodegradable and biobased polymeric materials in the form of 3D filaments based on polylactic acid and polyhydroxybutyrate suitable for additive manufacturing (AM).

Additive manufacturing, also known as three-dimensional (3D) printing, is an additive automated manufacturing process that uses digital 3D designs to prepare specific three-dimensional objects. In fact, it is a set of methods based on layered or spatial storage technologies that use computer-aided design (CAD) software to create physical 3D objects. These production methods allow the direct production of shape-complex products with precisely defined shapes [5].

Three-dimensional (3D) printing is a relatively new technique which dates back to 1980, but it is evolving dynamically. It includes the production of 3D spatial objects based on digital models. Currently, synthetic and biodegradable materials are both used for 3D printing. The range of polymers used in additive manufacturing includes thermoplastics [6], thermosets [7], elastomers [8], hydrogels [9], functional polymers, polymer blends [10], composites [11], and biological systems [12], which have already enabled the creation of value-added materials. Metals or ceramics are also used in 3D printing, and a combination of different types of fillers is currently being researched [13,14,15]. The gradually growing interest by industries as well as the general public in additive production is reflected in increasing investments in this area [16]. Additive manufacturing can increasingly compete with traditional production processes, such as injection molding in the case of plastics or CNC processing in the case of metals.

The most well-known 3D printing technology is fused deposition modeling (FDM), which is widely used in research and many industrial sectors. FDM is used to create models, prototypes, as well as end products. It is an inexpensive and simple way of producing prototypes or products, and it is often used to produce complex products. Fused deposition modeling (FDM) is a widely used 3D printing technology for polymer and composite filaments due to its flexible and fast printing process, low cost, variety of materials, and high strength as well as toughness of materials [17]. Several successful applications of FDM technology have been published in the literature associated with various industries, such as aerospace engineering [18], the automotive industry [19], and bioengineering [20,21,22,23]. Three-dimensional printing is considered to be an environmentally friendly processing technology because its use reduces the generation of plastic waste and also, theoretically, it can use recycled polymers. These benefits have the potential to increase resource efficiency in production by reducing waste production, which contributes to the development of a circular economy (CE) system [24,25]. Among the polymers used in FDM, the most commonly reported polymers are polylactic acid (PLA) and acrylonitrile-butadiene styrene (ABS) copolymer [26].

There are several reasons why the development and use of bioplastics have recently gained significant attention in FDM technology. These reasons include environmental pollution caused by conventional synthetic plastic materials, a reduction in crude oil resources, and interest in controlling toxic gas emissions. The term “bioplastic”, in general, refers to polymer materials that are either biobased or biodegradable, or both [27,28,29,30].

Biodegradable materials from renewable sources contribute to sustainable development as well as reduce the negative impacts of polymeric materials on the environment. The main advantage of biodegradable materials from renewable sources is their biodegradability in a biologically active environment, and therefore their contribution to reducing the carbon footprint [2,31,32]. This means that additive production using biomaterials can contribute to many areas of a circular economy, such as saving input materials as well as their recycling and reuse [33,34]. FDM technology is also widely used in medicine and tissue engineering, for example, for the development of customized scaffolds for a wide range of biomaterials [22,35,36]. In addition, FDM technology can generate biomimetic scaffolds due to its ability to produce multi-material structures and push soft biomaterials such as hydrogels [37,38,39]. Despite the clear advantages of biopolymers obtained from renewable raw materials, some properties of these materials are inferior to those of commonly used plastics, in particular, thermal and mechanical properties, as well as processing stability. Therefore, scientists are trying to find bioplastics that have comparable or even better properties than synthetic (petroleum-based) plastics. Biodegradable materials from renewable sources (biopolymers) used in FDM technology include polylactic acid (PLA), mainly due to its excellent biocompatibility, good biodegradability, and high mechanical strength [40,41]. Polylactic acid is a transparent, colorless, and solid thermoplastic material. It is synthesized from lactic acid (its dilactide), which is produced by fermentation of renewable raw materials. PLA is a chiral compound, which gives it various rheological, mechanical, and thermal properties. Polylactic acid may be a suitable replacement for PP (polypropylene), PS (polystyrene), or ABS (acrylonitrile butadiene styrene) in industrial applications [42,43,44,45]. PLA has been evaluated as the most suitable material for 3D printing by the FDM method because it requires a relatively low print temperature (low die and pad temperature) and also has excellent adhesion to the 3D printer pad [46]. However, this material also has certain drawbacks, in particular, very high brittleness and poor shape stability at temperatures above 60–70 °C. In addition, PLA is sensitive to thermomechanical and hydrolytic degradation during thermomechanical processing [47], with cleavage of the polymer chain, which causes a decrease in molecular weight and inner viscosity. During its use, the polymer may be sensitive to thermal, photochemical, and hydrolytic degradation, which also contributes to reducing its molecular weight [48,49]. Its shortcomings also include the fact that it is not biodegradable in home compost. For these reasons, the possibility of modifying polylactic acid in order to minimize its undesirable properties has been investigated [45].

One possible modification to improve the insufficient properties of PLA is to prepare polymer blends with another biopolymer. An alternative is the use of polyhydroxyalkanoates, which are obtained from renewable raw materials and are fully biodegradable and compostable. Polyhydroxyalkanoates (PHA) are a class of biopolyesters that can be produced naturally by bacteria. The most commonly used type of PHA is poly (3-hydroxybutyrate). Due to its biocompatibility and biodegradability, PHB has attracted attention especially for its possible applications in biomedical areas, for example, as scaffolds for tissue engineering. PHB is a thermoplastic polyester classified as a biopolymer that is produced by microorganisms such as *Ralstonia eutropha* under non-equilibrium growth conditions [50,51]. It is a semi-crystalline polyester with a high level of crystallinity. Although PHB is a fully biodegradable biopolymer, when used alone, it is relatively brittle, expensive, and sensitive to thermomechanical degradation during processing [2]. It is usually used in polymer blends to modify the resulting properties instead of applying it in pure form [2,52,53]. Several authors have suggested blending PLA and PHB as a strategy to address deficient material characteristics and improve the properties of the original polymer [29,53,54,55,56]. Previous studies of PLA/PHB blends have shown that the miscibility between the two polymers depended on the molecular weight of the second component [57,58,59]. The properties of PLA/PHB blends have also been shown to depend on the composition, chemical or physical crosslinking, and processing conditions. PLA/PHB blends alone show relatively low values of flexibility [29,60]. To overcome this disadvantage, it has been proposed to incorporate plasticizers into a PLA/PHB blend, which leads to a reduction in the glass transition temperature (T_g_), better processability of the material, and improved flexibility [61,62,63]. The EP2710076B1 patent [64] shows that blending PLA and PHB and a suitable type of plasticizer improves the mechanical properties and even increases the flexibility of the PLA/PHB blend. The most significant prerequisite for using these materials is the production of cast and blown films, but it has been found that they are also suitable for use in FDM 3D printing technology [65,66]. Studies by [45,65] examined the printability of PLA/PHB blends using various laboratory-prepared and commercial-type plasticizers, with improved dimensional stability and warping reduction by using 15 wt% plasticizer, as well as a positive effect on printability itself and improved physical and mechanical properties (especially impact strength and elongation at break) as compared with PLA/PHB blends without plasticizers and also as compared with PLA itself. Another study examined the effect of temperature and printing conditions of extruded PLA/PHB filament [67]. The contact time as well as the surface contact area were found to affect the crystalline phase of the blends and their subsequent biodegradability.

Since PLA/PHB blends have recently been widely researched in the field of additive production and also in other branches of the plastics industry due to their positive properties, it is interesting to examine the possibility of their material recycling, especially in terms of a circular economy and environmental protection. Researchers [68,69,70,71] have emphasized that a mechanical recycling strategy is the most appropriate plastic waste management option for the recovery of relatively clean and homogeneous plastic waste as compared with landfill or incineration alternatives. Mechanical recycling allows direct recovery of solid plastic waste for re-use in the production of new plastic products [72,73,74] and can use traditional technologies and commonly used processing machines such as conventional extruders and injection molding machines. Due to the fact that the recycling process results in a polymer with varying degrees of degradation, mechanical recycling is limited by the number of repeated processing cycles [75]. Material recycling has commonly been used for many years in the management of plastic waste from the industrial processing of polymeric materials [76,77]. According to the European Strategy for Plastics in the Circulating Economy [78], by 2030, all plastic packaging placed on the EU market should be either reusable or recyclable in a cost-effective way and more than half of all plastic waste generated in Europe should be recycled. According to this strategy [78], innovative materials and alternative starting materials for plastics production should also be developed and used if the evidence clearly shows that they are more sustainable as compared with non-renewable alternatives.

Polymer degradation in an extruder is one of the main factors that can adversely affect the quality of plastic material during its recycling. The term deterioration of a material means deterioration of processability, the mechanical or thermal material properties, as well as the electrical and optical properties, or the appearance and aesthetic properties of the material [79,80]. Due to these effects, recycled plastics are usually used for the production of low technical value products (down cycling). Polymer degradation during processing also reduces the productivity of the reused material due to differences in the viscosities of the recycled and virgin polymers when they are blended to obtain the respective products. Degradation is affected by many factors, such as shear stress, temperature, humidity, pH, the presence of oxygen, UV radiation, and the presence of other technological additives (such as stabilizers) [81,82]. The growing amount of plastic waste produced, legislative requirements, and the increased interest of people in protecting the environment are reasons for the use of plastics in the largest possible number of “life” cycles. Therefore, it is very important to recognize the effect of multiple extrusions of different polymeric materials on the final properties of these materials. However, so far, only a small number of scientific articles have been devoted to experimentally describing changes in the properties of biodegradable types of renewable materials after multiple treatments of these materials [83,84], most of which have dealt with the reprocessing of PLA itself.

Multiple extrusions of PLA (10 processing cycles) were investigated in [85]. Samples were prepared using a twin-screw extruder. In this case, the tensile strength values of PLA did not depend significantly on the number of extrusion cycles. Another study [86] on PLA reprocessing showed that with multiple thermomechanical processing only the tensile modulus remained constant for up to seven injection cycles. Other tested mechanical properties as well as rheological properties decreased by multiple processing by injection. The viscosity of the PLA obtained using a shear rate gradient from 0.01 s^−1^ to 100 s^−1^ decreased significantly (from 3960 to 713 Pa) after the first processing cycle. The mechanical properties of recycled PLA after multiple treatments have been reported to become insufficient for subsequent industrial use. In [87], the influence of multiple thermomechanical processing on the structure and properties of an amorphous PLA type was studied. As a result of multiple processing of amorphous PLA, there was a decrease in molecular weight and, subsequently, a decrease in other tested properties of PLA after the second processing cycle. In [88], it was found that the effect of multiple extrusion of PLA led mainly to a reduction in elongation at break and also to a reduction in impact strength. The authors of [89] studied the effect of multiple processing on the properties of PHB itself and were able to regranulate PHB in three cycles. During the material recycling of PHB, its mechanical properties deteriorated significantly when the tensile strength of the material after triple extrusion reached less than 50% of the original tensile strength value.

Material recycling of PLA/PHB blends with the addition of a suitable plasticizer was tested in [2]. It was found that the molecular weight as well as the viscosity of the tested materials decreased due to multiple treatments. Nevertheless, after repeated processing, there was no decrease in the strength characteristics or a change in the thermal properties of the blends. It was proven in this work that it was possible to implement up to 11 processing cycles from PLA/PHB polymer blends with the addition of a suitable plasticizer. Despite the fact that PLA/PHB polymer blends are completely biodegradable in a suitable biological environment, in order to save raw material resources, it is important to know their material recycling potential and the associated reuse before such materials are composted. Such a procedure would significantly increase the ecological and economic value of using polymeric materials based on PLA/PHB, which directly corresponds to the principles of the circular economy. 

In this article, we focus on the effect of multiple processing on the final mechanical and thermal properties, as well as the processing stability of a PLA/PHB blend intended for 3D printing under the designation NONOILEN^®^ 3D 3056-2, which was processed in both laboratory and industrial conditions at an industrial partner. 

## 2. Experimental Methods

### 2.1. Materials and Methods of Samples’ Preparation

This work deals with the study of the recyclability of a biodegradable polymer blend intended for 3D printing technology under the trade name NONOILEN^®^ 3D 3056-2, which is a biodegradable polymeric material prepared from polylactic acid, polyhydroxybutyrate, and oligomer plasticizer. The concentration of PLA in the tested polymer blend is above 10%, the concentration of PHB is above 30%, and the concentration of suitable plasticizer is less than 15%. The exact composition of the blend was provided by Panara a.s., producer of NONOILEN^®^ 3D 3056-2. This type of blend is suitable for FDM processing by 3D printing technology, and was used as a basis for examining the possibility of multiple processing of the material with the possibility of its subsequent material recycling.

The process of material recycling (multiple extrusion) of the studied biodegradable polymer blend was carried out in two different processing streams.

For the study of experimental material recycling (under laboratory conditions), first, the material recycling of the tested blend was carried out under laboratory conditions at the Institute of Natural and Synthetic Polymers, Slovak University of Technology, in Bratislava, where the material was blended on a laboratory twin-screw extruder from the Labtech company and the resulting material was cyclically processed on a single-screw extruder (Plasticorder, Brabender, Duisburg, Germany). 

#### 2.1.1. Polymer Blend Preparation for the Study of Experimental Recycling

The polymer blend NONOILEN^®^ 3D 3056-2 for experimental recycling was prepared according to the recipe provided by Panara a.s. using a laboratory twin-screw extruder LTE16 from the Labtech company. The diameters of the screws were 16 mm and the L/D ratio was 40. The geometric arrangement of the screws was co-rotating with intermeshing. The screw speed was set at 150 min^−1^. The operating temperature profile of the device in the direction of the hopper to the head which consisted of 10 heating zones was as follows: 60–140–5 × 190–160–150–150 °C.

#### 2.1.2. Multiple Extrusion of the NONOILEN^®^ 3D 3056-2 Experimental Samples

The tested polymer blend based on PLA/PHB was processed several times on a Plasticorder Brabender single-screw extruder with a nozzle with a circular cross-section and a diameter of 2 mm. The screw speed was set at 55 min^−1^. The total retention time of the material in the device was 58 s. The temperature profile of the device in the direction of the hopper to the head was as follows: 170–185–190–180 °C. The granulate was fed into the device via a hopper. The melt was forced through a die into a water bath where the filaments were cooled, and then granulated. In total, the material was processed 9 times.

#### 2.1.3. Study of Industrial Material Recycling (in Real Conditions)

In the second processing stream, the blend with the same composition was prepared and further recycled under industrial conditions at the Fillamentum Manufacturing Czech company, Hulín, Czech Republic using a twin-screw extruder LTE 26 from the Labtech company, Praksa, Thailand. The screw diameter was 26 mm and the L/D ratio was 40. The geometric arrangement of the screws was co-rotating with intermeshing. The screw speed was set at 400 min^−1^ and the material was repeatedly processed, firstly by extrusion into filaments, and then by FDM technology (3D printing) into test specimens. Then, the prepared 3D specimens were mechanically ground into granules, from which the filaments were prepared in the next step.

#### 2.1.4. Polymer Blend Preparation for the Study of Material Recycling in Industrial Conditions

The polymer blend NONOILEN^®^ 3D 3056-2 for recycling in real conditions was provided by the Fillamentum Manufacturing Czech Co. as a commercially available material in the form of filaments suitable for 3D printing using FDM technology. Then, the provided filaments were processed using a 3D printer and the printed samples were mechanically granulated and reprocessed by extrusion again to the filament form on a Plasticorder Brabender single-screw extruder. In the case of studying the possibility of material recycling of the tested blend in real conditions, this blend was repeatedly thermomechanically stressed by a combination of processing using a single-screw extruder (filament production) and processing using a 3D printer (preparation of 3D samples).

### 2.2. Multiple Extrusion of the NONOILEN^®^ 3D 3056-2 Industrial Samples

The samples were prepared using a Plasticorder Brabender single-screw extruder with a die with a circular cross-section. The temperature profile used in the direction from the hopper to the head was as follows: 170–185–190–180 °C. The granules were fed into the device via a hopper. The melt was extruded through a nozzle into a water bath, where the prepared filaments were cooled and wound on a spool. The diameters of the prepared filaments were 1.5 mm ± 0.05 mm.

### 2.3. Preparation of the Samples by 3D Printing from Industrial Samples of NONOILEN^®^ 3D 3056-2

Samples were prepared by 3D printing. Tensile bars were printed on a PRUSA i3MK3 3D printer, Praque, Czech Republic. The prepared filaments were introduce into a 3D printer, where they were melted, and then printed in a form of tensile bars. The diameter of the nozzle of the 3D printer was 0.4 mm and its temperature was 190 °C. The temperature of the printer pad was 20 °C. The height of the printed layer was 0.2 mm and number of the layers was 10. The printing speed was 40 mm/s. 

The process of mechanical recycling in experimental and industrial conditions is shown in Figure 1 and the designation of each studied sample is stated in Table 1.

## 3. Experiments

The experiments were performed to describe the effects of multiple processing of the tested biodegadable polymer blend on its rheological and utility properties, in laboratory (experimental) as well as real (industrial) conditions. The aim was to investigate the possibility of material recycling of the tested biodegradable polymer blend based on PLA and PHB. 

### 3.1. Processing (Thermomechanical) Stability 

Testing of the processing stability of the biodegradable polymer blend samples was performed on an RPA 2000 oscillating rheometer, Wilmington, DE, USA. The tested samples were measured at a temperature of 190 °C. The preheating time was 1 min and the total duration of the test was 10 min. The oscillation frequency was 50 CPM and the oscillation angle was 60°. The samples were tested at a shear rate of 22 s^−1^. The dependences of the complex viscosity on the test time were determined. For a better comparison of the results, since the samples did not have the same viscosity at the beginning of the test, the dependence of the relative complex viscosity on time was evaluated as the indicator of degradation. The mathematical formula for calculating the relative complex viscosity is $\eta_{rel.}^{*}\left(t\right)=\frac{\eta^{*}\left(t\right)}{\eta^{*}\left(0\right)}$, where $\eta_{rel.}^{*}\left(t\right)$ is the relative complex viscosity in time *t*, $\eta^{*}\left(t\right)$ is the complex viscosity in time *t*, and $\eta^{*}\left(0\right)$ is the complex viscosity at the beginning of the test [90].

It should be noted that testing of the effect of multiple processing on processing stability, thermal properties, and physical-mechanical properties was performed under the same conditions for samples of both recycling processing streams.

### 3.2. Molecular Characteristics

Molar mass of the tested material was determined by gel permeation chromatography. The measurements were performed using an Agilent Technologies 1100 Series, Santa Clara, CA, USA instrument equipped with an isocratic pump, and an autosampler. A PLgel 5 μm mixed C column was tempered to 30 °C with chloroform as the eluent at a flow rate of 1 mL/min. Linear polystyrene standards with narrow distribution were used to gain the calibration curve (10 points in calibration). The instrument was equipped with a refractive index detector. 

### 3.3. Color Changes Measurement

Color changes was measured on prepared 3D printed samples according to the CIE Lab color scale relative to the standard illuminant D65 over a white background using a reflection spectrophotometer (Techkon SpectroDens, Königstein, Germany ). The aperture diameter of the measuring port was 3 mm. The selected illuminating and viewing configuration of this instrument was CIE diffuse/10° geometry and the CIE 1964 supplementary standard colorimetric observer. Total chromatic change was calculated using Equation (1):(1)$$\Delta E_{a,b}^{*}=\sqrt{{\left(\Delta L^{*}\right)}^{2}+{\left(\Delta a^{*}\right)}^{2}+{\left(\Delta b^{*}\right)}^{2}}$$

### 3.4. Measurement of Thermal Properties 

The thermal properties of the studied blend were measured using a differential scanning calorimetry on a Mettler-Toledo Inc., Worthington, OH, USA instrument. The basic thermal characteristics such as the glass transition temperature, the crystallization temperature, and the melting point of the crystallites were evaluated from the measurements. The conditions for measuring of thermal properties are summarized in Table 2.

The heat difference required for heating by 1 K between the sample and the standard was measured. The standard in this case was an empty cup. Nitrogen was used as the inert gas and the evaluation software was SW STARe 16.30.

### 3.5. Measurement of Mechanical Properties 

The measurement of mechanical properties was performed using a Zwick/Roell, Ulm, Germany testing device at a cross-head speed of 50 mm/min. The distance between the heads of the testing device was 50 mm and the working distance of the extensiometer was 20 mm. From the obtained tensile curves, the following mechanical parameters were determined: relative elongation at break (*ε_b_*), tensile strength at break (*σ_b_*), yield strength (*σ_y_*), and Young’s modulus (*E*). Mechanical properties were measured from prepared filaments and also from dogbone-shaped samples.

### 3.6. Flexural Test

The three-point bending test was performed on a UMZ-3K tester from the MICRO-EPSILON, Ortenburg, Germany company. The tested samples were prepared in the form of a filament with a circular cross-section as part of the material recycling testing under industrial conditions. The distance of the extreme support points was set to 24 mm in this test. The load was applied using a load pin in the middle of the span length with a cross head speed of 2 mm/min. During the test, the stress-strain dependency of the test specimen was recorded.

Flexural strength was evaluated by Equation (2):(2)$$\sigma_{max}=\frac{8\cdot F_{max}\cdot g}{\pi \cdot d^{3}}$$
where *F_max_* is the maximal developed force during the flexural test, *d* is the filaments diameter, and *g* is the distance between supports

Young’s modulus was evaluated as the slope of the first linear function area of dependency of flexural strength on sample deformation. 

Deformation at flexural strength was evaluated like a cross point of the tangents line of the first and second linear function area of dependency of flexural strength on sample deformation.

## 4. Results

### 4.1. Processing Stability

Figure 2 shows the dependence of complex viscosity on the material time in the oscillation field of the rheometer of individual processing cycles of the PLA/PHB blend NONOILEN^®^ 3D 3056-2 from Panara, a.s., Nitra, Slovak Republic company recycled in laboratory conditions. It can be seen from the graph that due to the longer retention of the material in the oscillating rheometer, the complex viscosity decreases, which is a demonstration of material degradation under thermomechanical loading. A decrease in complex viscosity also occurs due to multiple processing of the polymeric material. From this, it can be deduced that the investigated material degrades under thermomechanical loading in laboratory conditions. Degradation of recycled material under industrial conditions also occurs as a result of multiple treatments, as is clearly seen in Figure 3. Degradation of the material preferably occurs during filament processing. The largest decrease in complex viscosity during the test of thermomechanical stability is exhibited by Sample 2(I), but the largest decrease in the values of complex viscosity between two identical products processed with higher processing number occurs between the preparation of the first and the second filament (extrusion number 2(I) vs. 4(I)). On the contrary, the processing of the material using FDM 3D printing technology does not have a negative impact on the processing stability of the tested material. Since the experimental (laboratory) and industrial processing equipment differ from each other, both in terms of construction and also in different retention time of the material during processing, to compare the results between commercial and laboratroy tested material, we chose extrusion times as the comparison parameter. The total processing time of the material for each processing pass of the tested polymeric blend is given in Table 1. Figure 4 shows a comparison of the complex viscosity of an experimental sample and a sample processed under real (industrial) processing conditions. During the blending under industrial conditions, the material is thermomechanically loaded by different processing conditions as compared with blending of the experimental polymer blend (different shear stress of the material during blending, different time of retention), which causes a higher rate of material degradation during blending as compared with material blending under laboratory conditions.

### 4.2. Molecular Characteristics

To better understand the problem of degradation of the studied biodegradable polymer blend based on PLA and PHB, we performed measurements of molecular characteristics of individual processing cycles of the polymer blend, which were processed in experimental, and subsequently, in real production conditions. Figure 5 shows the molar mass distributions for the individual processing cycles of the material tested in both processing conditions. In both cases, due to multiple treatments, the molar masses shift to lower values, which is due to the degradation of the material during the multiple thermomechanical loading. The biggest difference between the molar mass distributions in laboratory (experimental) processing of the tested material and processing in a commercial processing machine is the absence of macromolecules with the highest molar mass ($\bar{M_{w}}$ above 10^6^ Da) in the case of industrial processing. This result is consistent with the results described in the processing stability test, where the commercially processed material showed lower complex viscosity values than the material processed under laboratory conditions. 

The different influences of processing on the material’s molecular characteristics are clearly shown in Figure 6; the dependence of the molar mass on the retention time of the material in the processing equipment is shown, where a more significant decrease in the molar mass occurs during the experimental processing of the studied material. In both processing alternatives, however, the longer total processing time causes a decrease in the average molar mass. There is a very good correlation of results between the processing stability of the tested materials and their average molar mass (Figure 7), both in the case of laboratory and industrial samples of individual processing passes. 

In the case of the polydispersity index (Figure 8), the situation is different. Due to the absence of large macromolecules ($\bar{M_{w}}$ above 10^6^ Da) in the case of blend samples processed under industrial conditions, these samples show a narrower distribution curve than in the case of blend samples processed under laboratory conditions. In the case of laboratory-processed samples, a decrease in polydispersity is visible due to multiple processing of the material, and thus, longer retention of the material in the processing equipment. During industrial processing of the material, the polydispersity index changes only minimally due to multiple processing, because the main molar mass reduction of the highest molar mass fractions was run during the blend preparation. The lower polydispersity index values for samples processed during industrial recycling was caused by the fact that, in the case of blending the polymer blend under industrial conditions, a twin-screw extruder with a higher processing time and a higher shear stress was used as compared with the laboratory blending equipment. Macromolecules with lower $\bar{M_{w}}$ are more stable against thermomechanical degradation during processing because of lower viscosity and lower shear stresses in the melt.

From the study of material degradation during multiple processing in machines, we found that, due to multiple thermomechanical processing, the tested material partially degrades, which was associated with a decrease in its molar mass and a subsequent decrease in material viscosity. It should be noted that the starting blends themselves differed in viscosity. Greater degradation of the material occurs during its processing under industrial conditions, which is due to the different construction of industrial equipment, different shear stress of the material, as well as different retention time of the material in the processing equipment. However, the positive fact is that both in the laboratory, and subsequently in the research phase directly at the industrial partner, we were able to realize nine processing passes, which would ultimately mean that the tested blend would be usable at least four times under industrial conditions before its ecological recovery in the form of biodegradation would be necessary.

### 4.3. Color Changes

As multiple material processing in the processing machines results in material degradation, which often results in darkening of the material, we examined the color changes accompanying material recycling. Figure 9 shows the measured tensile specimens after 3D printing of industrially processed material, where the first dogbone represents the third processing cycle, the second dogbone represents the fifth processing cycle, the third dogbone represents the seventh processing cycle, and the fourth dogbone represents the last, i.e., ninth processing cycle of the NONOILEN^®^ 3D 3056-2 material.

Figure 10 shows the dependence of the brightness of the material on the retention time of the material in the device for extruded 3D objects of industrially processed sample types and for filaments of the experimentally processed tested polymer blend. It can be seen that during recycling under laboratory conditions, due to the extrusion process, the brightness values decrease, and thus, the material darkens. In the case of samples prepared by 3D printing, the brightness values change minimally. It can also be seen from the dependence of the color difference of the material (Figure 11) on the time in the device that the color difference is larger as compared with the initial sample in the case of material processing under laboratory conditions; in industrial processing the color difference changes minimally.

Significant color change in samples processed under laboratory conditions occurs due to differential degradation of the primary non-recycled polymer blend during its blending when the blend prepared under industrial conditions already shows lower lightness values due to degradation during blending of the polymer blend. For this reason, in the process of multiple processing under industrial conditions, such a significant color change can no longer occur as in the case of processing samples prepared under laboratory conditions.

In the case of experimental and industrial processing of samples, different effects of processing equipment on processing stability, molecular characteristics, and color changes of the biodegradable polymer blend type NONOILEN^®^ 3D 3056-2 were found. In the following section, the effects of multiple processing on the thermal and physical-mechanical properties were tested to clarify the possibility of material recycling of the tested biodegradable polymeric material. The material was also processed first in laboratory conditions, and then in industrial (commercial) processing directly at the industrial partner.

### 4.4. Thermophysical Properties

The thermophysical properties were measured using the DSC method, with first heating of the samples, cooling, and then second heating (Figure 12, Figure 13 and Figure 14). For illustration, in these figures, thermograms are plotted for the first and ninth processing passes in the case of the experimental part of the study. In the case of mechanical recycling in industrial conditions, in the graph, thermograms are plotted for the first and the fourth filaments and for the first and last (fourth) printed 3D samples. The process of thermophysical measurements measured by DSC of other tested samples was comparable with the given thermograms. The thermophysical characteristics of the processing passes for the experimental and industrial samples are summarized in Table 3. Based on the evaluation of thermal properties of individual processing cycles prepared in laboratory and industrial conditions, we found that multiple processing of the tested blend had only a minimal effect on its thermophysical characteristics, regardless of the processing equipment on which the polymeric material was processed. By comparing the different technological equipment on which the materials were processed, only slight deviations were recorded at the values of the glass transition temperature (T_g_) depending on the used technology, while the samples prepared under laboratory conditions showed a glass transition temperature from 2 to 3.5 °C higher than in the case of samples prepared on industrial equipment (filaments and 3D samples). Due to multiple processing of the material, the values of the cold crystallization temperature and the melting point of the crystallites do not significantly change. In the case of cooling, all processing transitions of the tested material show a crystallization peak, where there is also no shift in the crystallization temperature values due to multiple processing. In addition, due to multiple treatments, the enthalpy of melting crystallites, crystallization, and cold crystallization does not change, despite the fact that multiple treatments cause degradation of the material accompanied by a decrease in molar mass. The results indicate that in both cases of multiple processing (in laboratory and industrial conditions) the ratio of crystalline and amorphous phase is without significant changes with multiple processing during recycling. However, the thermophysical properties are slightly influenced by the processing equipment as well as the blending process of the original (non-recycled) polymer blend, especially in the case of influencing the glass transition temperature and temperature of cold crystallization.

### 4.5. Mechanical Properties

Multiple processing during material recycling and the associated degradation of the material can also have an adverse effect on the mechanical properties of the studied polymeric material, while degradation often leads to a decrease in strength characteristics as well as a decrease in flexibility of polymeric material. Therefore, when measuring mechanical properties, basic physical-mechanical parameters such as yield strength, tensile strength, and relative elongation at break were evaluated. The mechanical properties of the tested processing cycles were evaluated separately for experimental samples and separately for commercial types of samples (filament, 3D object), while the values of mechanical properties were related to the total extrusion time. Figure 15 shows the dependence of the yield strength on the total retention time of the material in the processing machine. In the case of specimens prepared by 3D printing (3D samples), it was not possible to determine the yield strength of the material. By comparing the experimental samples and the commercial filament, which has been repeatedly treated, it showed that multiple treatments did not have a significantly negative effect on the yield strength. This effect could be predicted based on the unchanged amorphous phase/crystallinity ratio, as mentioned in comments about the DSC measurements. It can also be stated that multiple processing does not negatively affect the tensile strength (Figure 16) of the studied experimental and industrially processed types of samples. In the case of strength characteristics, no significant effect of the used processing equipment was recorded. In the case of relative elongation at break (Figure 17), it can also be seen that multiple processing under laboratory conditions has minimal effect on its values. On the contrary, processing in industrial conditions has a greater impact on the flexibility of the material. In this case, there was a decrease in the elongation at break after the preparation of the second filament. The decrease in elongation at break was due to the lower molar mass of these samples and the absence of long macromolecular chains during material blending on industrial equipment. Based on the fact that this type of blend is not intended for high flexible products (initial elongation at break is not so high), the changes in this parameter do not significantly devaluate the application properties of the final product. 

Scientific articles [91,92] state that, in addition to the decrease in molecular weight due to degradation, the deterioration in mechanical strength may also be caused due to the moisture content of polyesters, including PLA and PHB. In this case, due to multiple treatments, the strength characteristics were affected only minimally. That means that the possible different moisture contents of the samples after their multiple processing did not have a significant effect on the strength characteristics of the studied samples.

### 4.6. Flexural Test

Flexural strength provides information about the material’s ability to withstand deformation under stress. The flexural test was investigated on the filaments prepared in the process of industrial recycling. Figure 18 shows a slight decrease in the maximum flexural strength values of the individual samples due to multiple processing (recycling) of the tested polymeric material. The decrease in flexural strength values of the tested samples is caused by partial degradation during processing, when the molar mass of the polymer blend decreases, but the decrease in maximum flexural strength even after nine processing cycles of the polymer blend shows only 7% of the value as compared with the original non-recycled sample. In addition, only minimal changes during multiple processing occurred in the case of deformation at the flexural strength (Figure 19) and also in the evaluation of Young’s modulus (Figure 20), when the values of the investigated parameters were in the range of statistical measurement errors even after multiple processing as compared with non-recycled material.

During the 3D printing test, it was noted that multiple processing of material had no negative impact on the 3D printing. The tested material can be printed under the same conditions as fresh (unrecycled) material even after multiple processing (the same printing temperature, pad temperature, and printing speed). There were no increases in side effects such as warping, delamination of layers, or deformation of the 3D-printed model observed during the 3D printing of objects (samples) made of multiple processed material.

## 5. Conclusions

In this study, we investigated the possibility of material recycling of a biodegradable polymer blend based on PLA and PHB under the trade name NONOILEN prepared under laboratory and industrial conditions. Biodegradable polymers were tested in two different processing streams, namely in laboratory and industrial conditions, to compare the impact of multiple processing on processing stability, molecular characteristics, thermophysical properties, mechanical properties, and color changes of the tested materials. 

During the testing of processing stability of the studied material, it was found that the extended extrusion time in multiple processing caused degradation of the material due to cyclic thermomechanical loading, which was reflected in a decrease in the complex viscosity values of the tested material after multiple processing. A more significant decrease in complex viscosity due to multiple processing occurred in the polymeric material tested under laboratory conditions, especially at the third thermomechanical loading of the material (Sample 3(E)). In the case of testing the processing stability of the material processed under industrial conditions, the complex viscosity of the non-recycled material was lower as compared with the complex viscosity of material tested under laboratory (experimental) conditions, due to different types of multiple processing of material in laboratory and in industrial conditions. The highest decrease in viscosity, in the case of the industrial type of tested blend, was observed during the blending process of the polymer blend. Subsequently, a significant decrease in processing stability during the process of multiple processing (material recycling) under industrial conditions occurred during the fourth processing cycle (Sample 4(I)).

Degradation of the material was also proven by a GPC analysis, which showed that, in the case of industrial processing of this polymer blend, the degradation of long macromolecular segments occurred preferentially in the blending process of the biodegradable polymer blend. During multiple processing, a more pronounced decrease in the average molecular weight occurred, in the case of experimentally prepared samples, which had, however, a higher initial average molecular weight. The value of the average molecular weight stabilized in the range from 120 to 160 kDA, even after multiple processing, both in the case of experimentally processed samples and in the case of industrially processed samples. Nevertheless, it was possible to process the tested biodegradable material in both laboratory and industrial conditions in up to nine processing cycles, which, from an industrial point of view, means that after preparing the polymer blend it is possible to prepare the filaments and to use them to print an object in four consecutive recurring cycles.

It is positive that, even after partial degradation of the material due to multiple processing, there is no significant change in its thermophysical properties, either in laboratory or industrial conditions. The difference between thermophysical properties (mainly T_g_) of experimental and industrial samples was given by the different degradations of the tested material in the process of blending.

There is also no negative impact on the strength characteristics of multiple processed samples. These results correspond with results described in [2], where material recycling of PLA/PHB blends suitable for injection molding were tested. 

The results obtained in this work indicate that a biodegradable and biobased polymer blend based on PLA and PHB is a suitable candidate for material recycling even in industrial processing conditions. The largest differences in tested rheological and utility properties between the samples processed under experimental as well as industrial conditions were recorded during the actual blending of the biodegradable polymer blend on the different blending devices. The results of this scientific work suggest that the biodegradable polymeric material NONOILEN^®^ 3D 3056-2 is suitable for multiple uses in FDM technology, which could lead to significant savings in input materials in 3D printing, for example, in the event of defective prints.

## Figures and Tables

**Figure 1 polymers-14-01947-f001:**
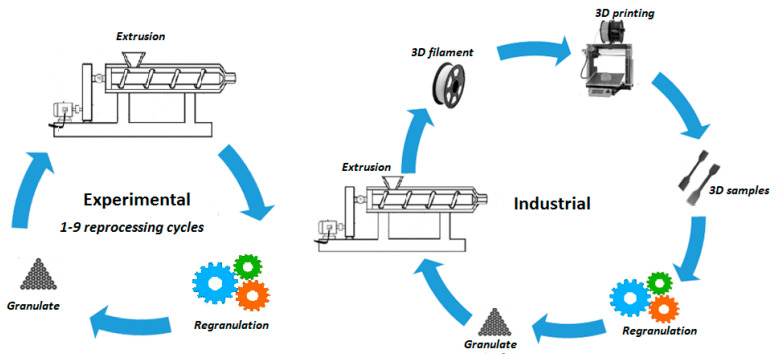
The process of mechanical recycling in experimental and industrial conditions.

**Figure 2 polymers-14-01947-f002:**
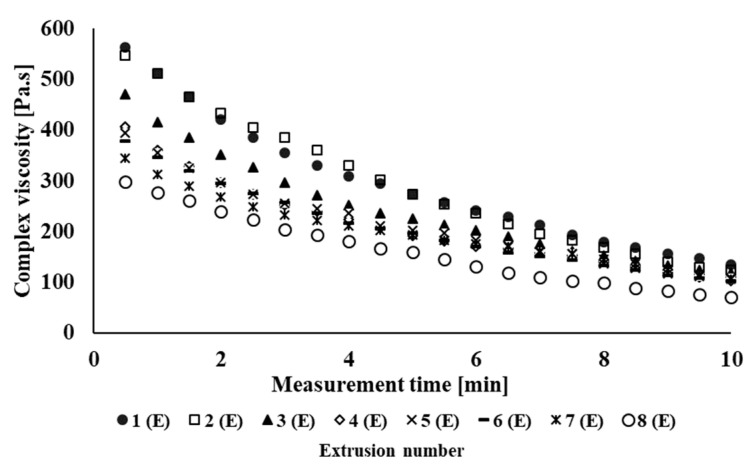
Dependence of complex viscosity on measurement time for NONOILEN^®^ 3D 3056-2 experimental samples prepared in laboratory conditions.

**Figure 3 polymers-14-01947-f003:**
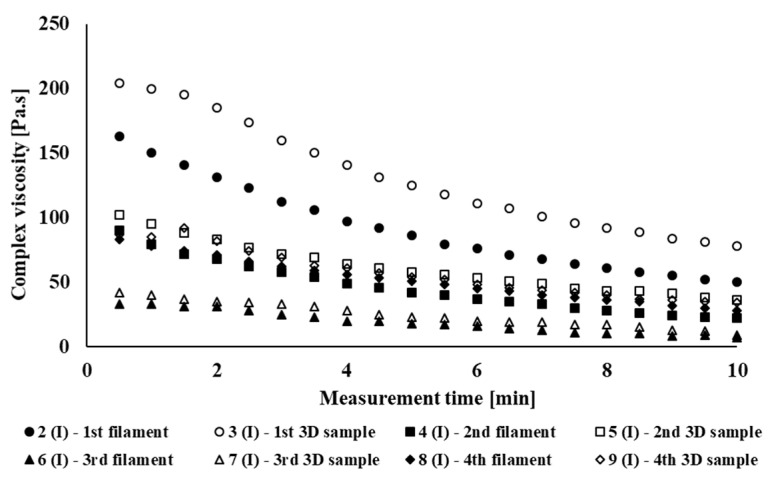
Dependence of complex viscosity on measurement time for commercially processed NONOILEN^®^ 3D 3056-2 samples.

**Figure 4 polymers-14-01947-f004:**
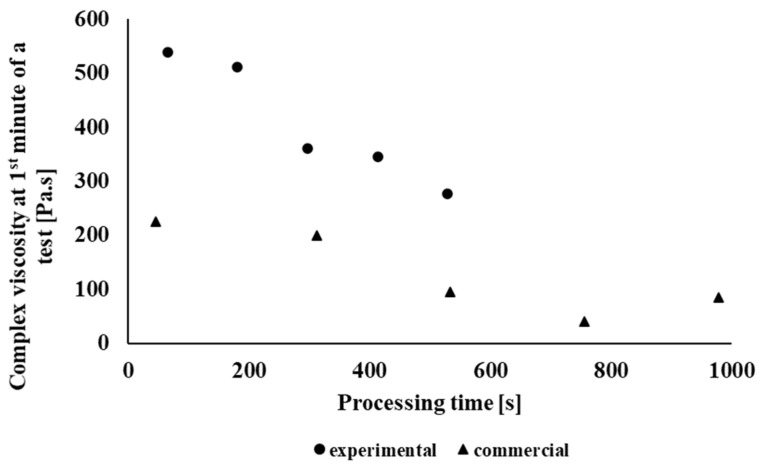
Dependence of the complex viscosity in the 1st minute of the test in an oscillating rheometer on the processing time of the NONOILEN^®^ 3D 3056-2 material.

**Figure 5 polymers-14-01947-f005:**
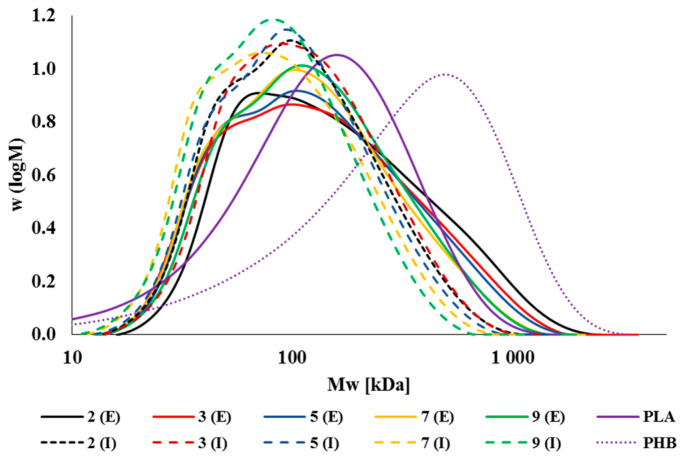
Molar mass distribution of individual processing cycles of experimental and industrial types of the NONOILEN^®^ 3D 3056-2 material.

**Figure 6 polymers-14-01947-f006:**
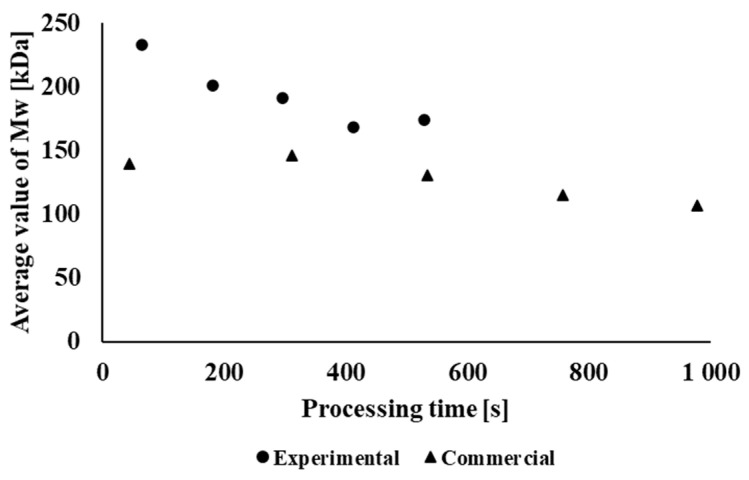
Dependence of the average molar mass as a function of the total retention time of the material in the processing equipment.

**Figure 7 polymers-14-01947-f007:**
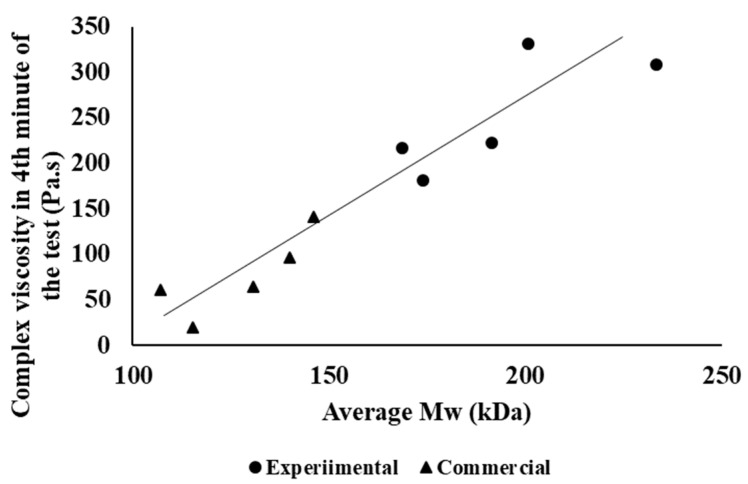
Dependence of complex viscosity in the 4th minute of the processing stability test on the average molar mass of the material.

**Figure 8 polymers-14-01947-f008:**
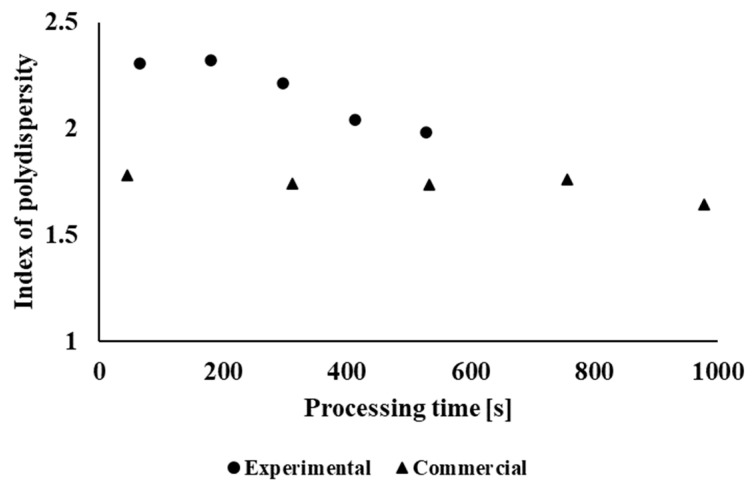
Dependence of the polydispersity index on the total retention time of the material in the processing equipment.

**Figure 9 polymers-14-01947-f009:**
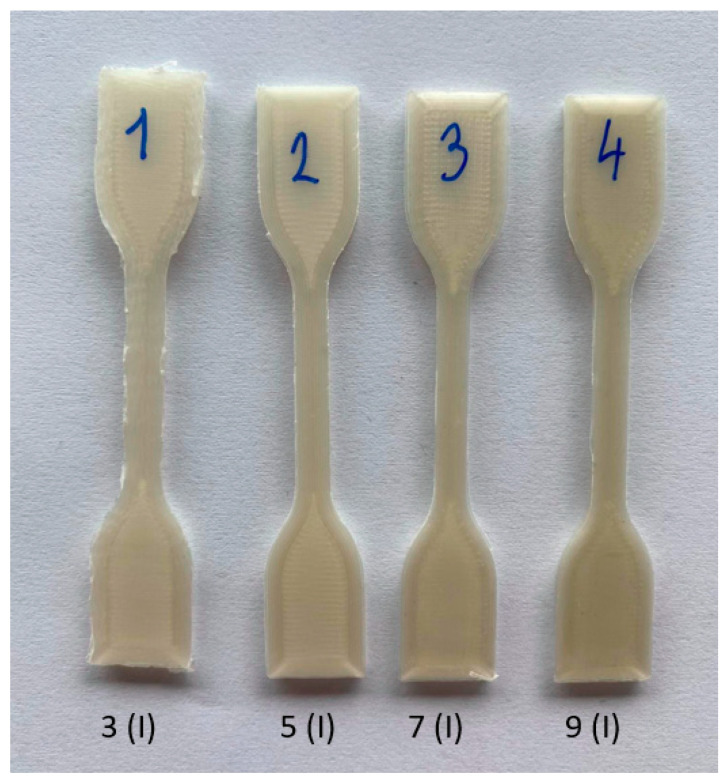
3D printed samples after multiple processing (3rd, 5th, 7th, and 9th processing transition) of the industrial type sample, NONOILEN^®^ Fillamentum for NATURE.

**Figure 10 polymers-14-01947-f010:**
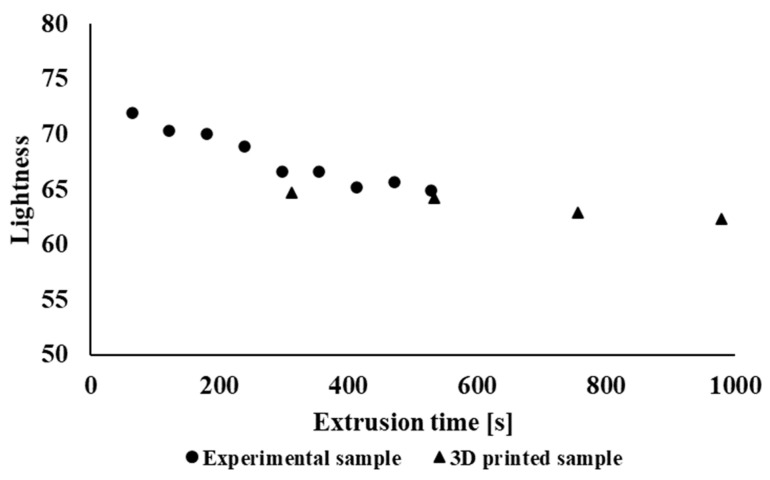
Dependence of material lightness on material retention time in equipment for experimental samples as well as 3D printed objects of industrially processed samples of NONOILEN^®^ 3D 3056-2.

**Figure 11 polymers-14-01947-f011:**
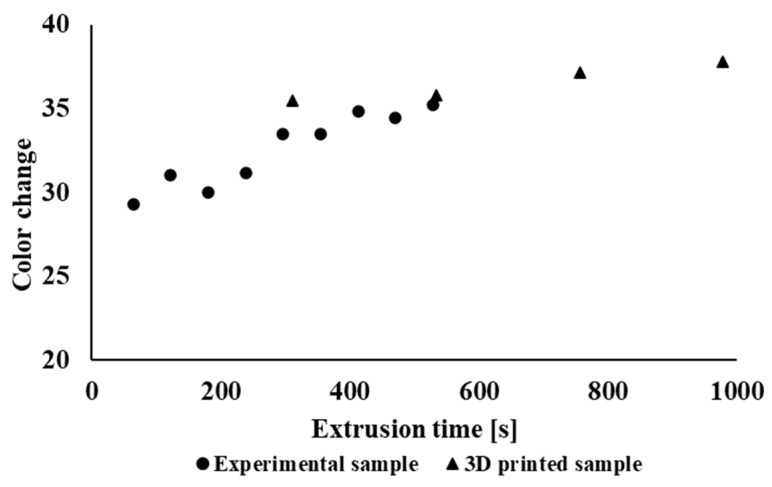
Dependence of the color difference of the material on the retention time of the material in the processing equipment.

**Figure 12 polymers-14-01947-f012:**
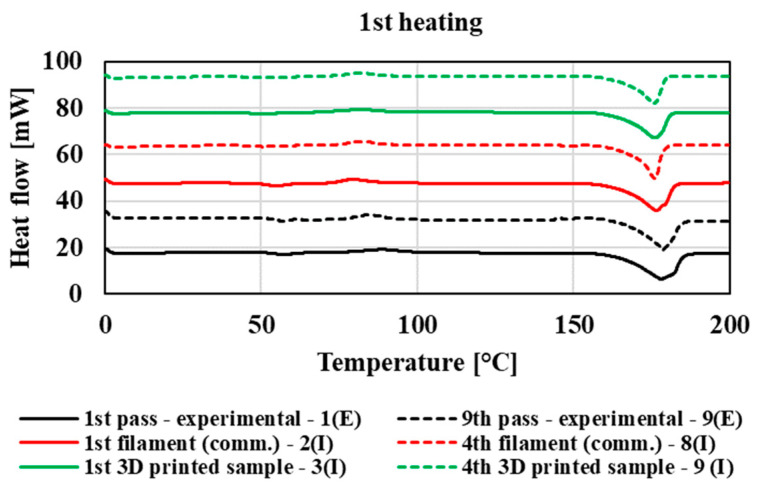
DSC thermogram of 1st heating for experimental and commercial type polymeric material.

**Figure 13 polymers-14-01947-f013:**
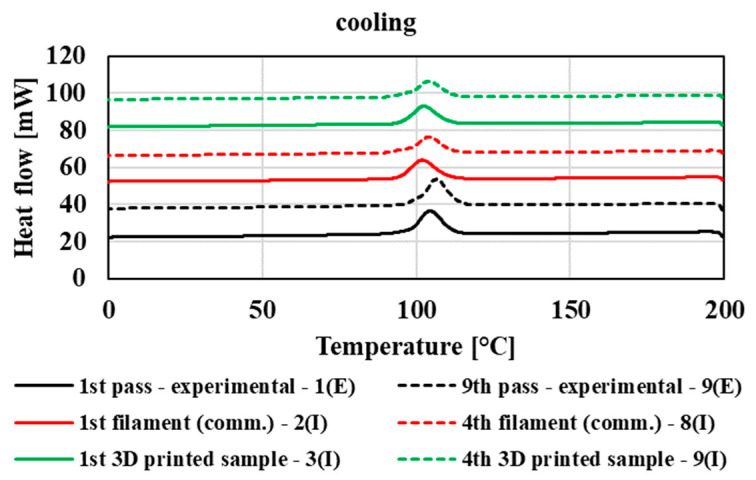
DSC thermogram of cooling heating for experimental and commercial type polymeric material.

**Figure 14 polymers-14-01947-f014:**
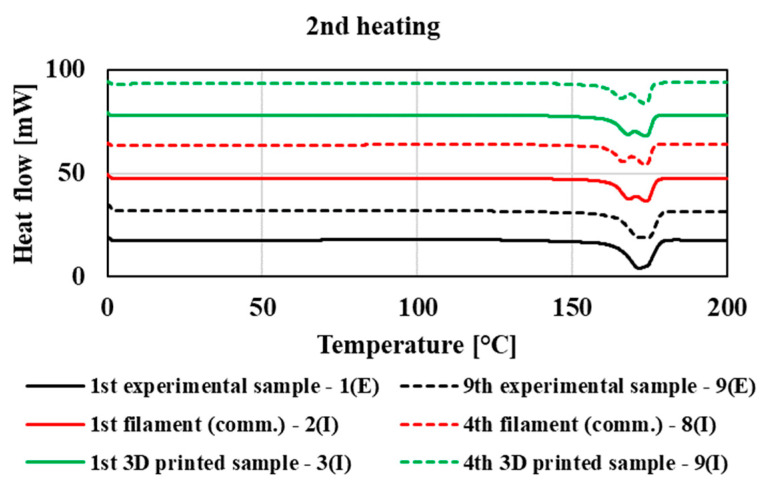
DSC thermogram of 2nd heating for experimental and commercial type polymeric material.

**Figure 15 polymers-14-01947-f015:**
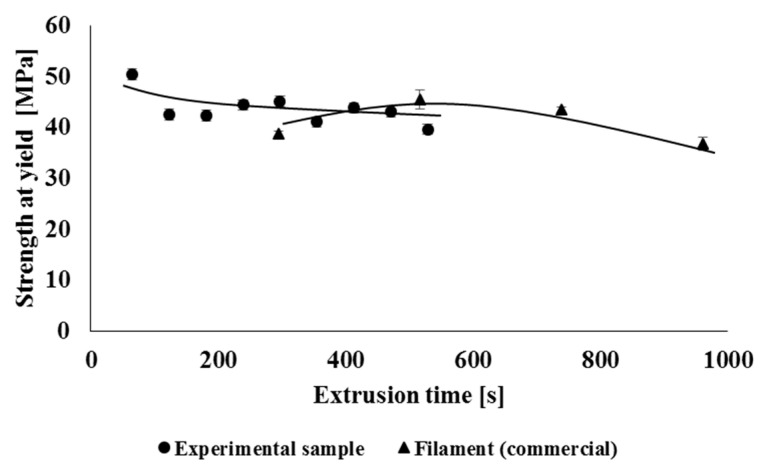
Dependence of strength at yield on the total processing time of the polymeric material.

**Figure 16 polymers-14-01947-f016:**
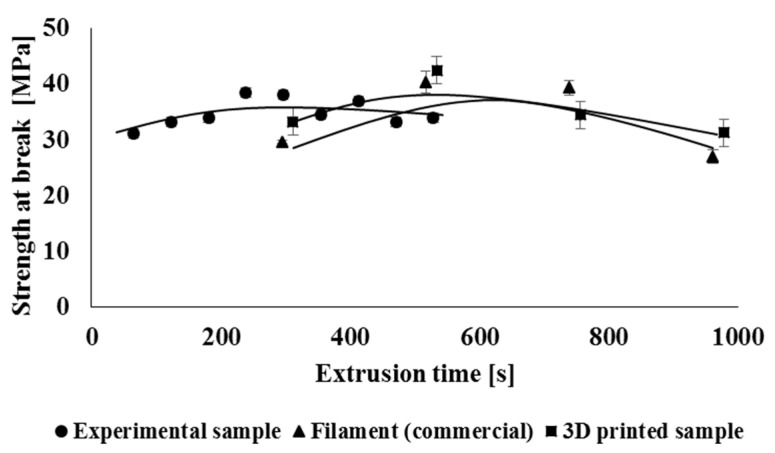
Dependence of tensile strength at break on the total processing time of the polymeric material.

**Figure 17 polymers-14-01947-f017:**
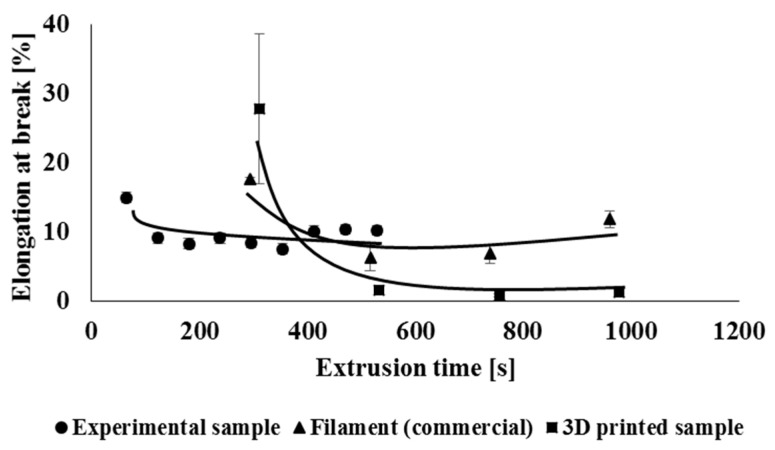
Dependence of the elongation at break on the total processing time of the polymeric material.

**Figure 18 polymers-14-01947-f018:**
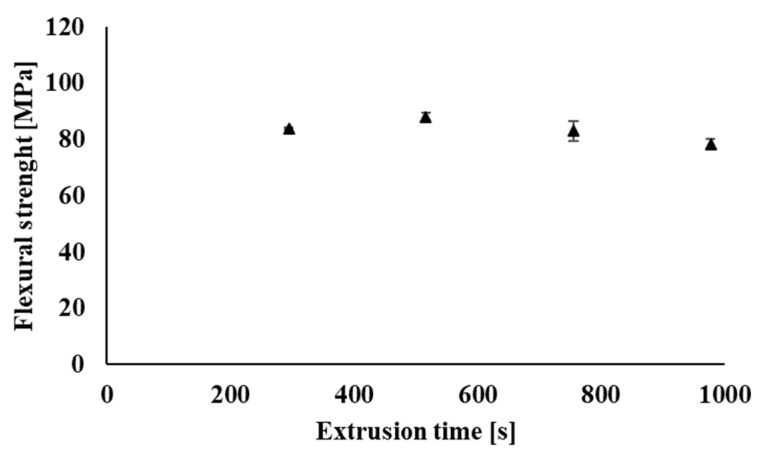
Dependence of maximum flexural strength on the total processing time of the material in the form of filaments processed under industrial conditions.

**Figure 19 polymers-14-01947-f019:**
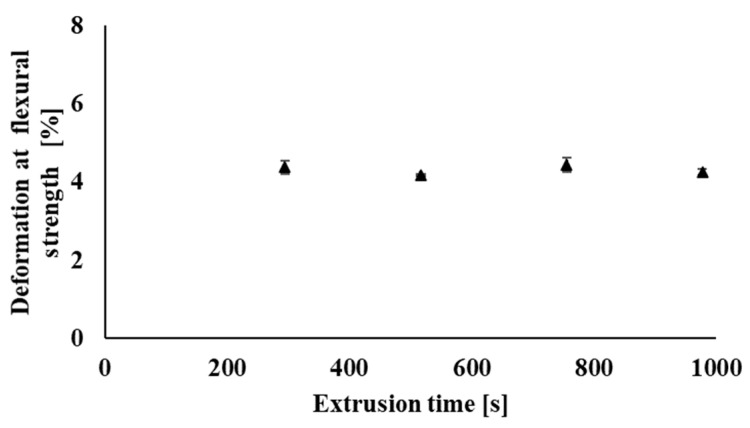
Dependence of flexural strength deformation on the total processing time of the material in the form of filaments processed under industrial.

**Figure 20 polymers-14-01947-f020:**
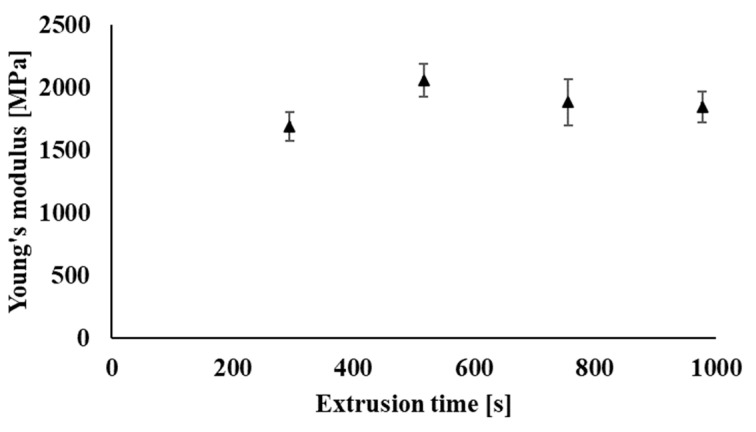
Dependence of Young’s modulus on the total processing time of the material in the form of filaments processed under industrial conditions.

**Table 1 polymers-14-01947-t001:** Processing passes of the PLA/PHB blend suitable for 3D printing.

**Experimental Recycling**			
**Extrusion Number/Sample Name**	**Processing Equipment**	**Processing Time (Seconds)**	
1 (E)	Twin-screw extruder (blending)	65	
2 (E)	Single-screw extruder	123	
3 (E)	Single-screw extruder	181	
4 (E)	Single-screw extruder	239	
5 (E)	Single-screw extruder	297	
6 (E)	Single-screw extruder	355	
7 (E)	Single-screw extruder	413	
8 (E)	Single-screw extruder	471	
9 (E)	Single-screw extruder	529	
**Industrial Recycling**			
**Extrusion Number/Sample Name**	**Processing Equipment**	**Processing Time (Seconds)**	**Product**
1 (I)	Twin-screw extruder (blending) *	45	Polymer blend
2 (I)	Single-screw extruder *	295	1st Filament
3 (I)	3D printer	312	1st 3D sample
4 (I)	Single-screw extruder	517	2nd Filament
5 (I)	3D printer	534	2nd 3D sample
6 (I)	Single-screw extruder	739	3rd Filament
7 (I)	3D printer	756	3rd 3D sample
8 (I)	Single-screw extruder	961	4th Filament
9 (I)	3D printer	978	4th 3D sample

* 1st and 2nd processing passes of the polymer blend were prepared in cooperation with the Fillamentum Manufacuring Czech company at their industrial processing facilities. Note: The symbol after the processing cycle number indicates how the material was recycled (I—in industrial conditions; E—in laboratory conditions).

**Table 2 polymers-14-01947-t002:** Conditions for measuring thermal properties.

	Ramp	Set Temperature (°C)	Time (min)
Conditioning	isothermal	0	1
1st Heating	10 °C/min	200	20
Conditioning	isothermal	200	3
Cooling	10 °C/min	0	20
Conditioning	isothermal	0	3
2nd Heating	10 °C/min	200	20

**Table 3 polymers-14-01947-t003:** Thermophysical characteristics of studied samples (processing passes).

Sample	Experimental Samples					Industrial Samples				
	1 (E)	3 (E)	5 (E)	7 (E)	9 (E)	1 (I)	3 (I)	5 (I)	7 (I)	9 (I)
T_g_ [°C]	53.76	54.75	54.67	54.73	54.74	48.58	46.75	47.76	49.11	48.28
T_cc_ [°C]	88.80	84.80	84.00	84.33	84.99	79.31	82.47	83.15	83.67	82.00
ΔH_cc_ [J/g]	6.46	9.56	9.38	8.16	8.70	5.90	4.95	7.03	6.20	6.65
T_m_ [°C]	177.36	177.09	178.26	177.60	179.78	175.84	176.08	177.38	175.09	175.06
ΔH_m_ [J/g]	56.75	53.93	53.57	53.30	50.28	57.70	55.46	55.17	53.99	53.92
T_cc_ [°C]	105.12	106.26	107.19	107.50	107.36	102.52	102.65	104.01	105.32	104.36
ΔH_c_ [J/g]	46.06	45.45	47.23	45.52	46.69	46.57	43.33	44.53	41.31	42.83

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
