# Peer review of "Influence of Multiple Thermomechanical Processing of 3D Filaments Based on Polylactic Acid and Polyhydroxybutyrate on Their Rheological and Utility Properties"

_polymers, 2022, doi:10.3390/polym14101947_

Round 1
Reviewer 1 Report
In this manuscript, the authors investigate the polymeric material recycling of the biodegradable blend based on PLA and PHB under multiple processing procedures and conditions. The results show that the properties of PLA and PHB can be maintained when being processed in the laboratory as well as in industrial conditions, with no negative impact on the properties of multiple processed blends. Accordingly, the authors claim the biodegradable polymeric material based on PLA and PHB is very suitable for its multiple uses in FDM technology, even in scale-up industrial processing conditions. The investigation and results are quite detailed and interesting. I would like to recommend a minor revision before the manuscript can be accepted.
Comments
1) Line 48, please give the full name of the abbreviations (ABS, PLA, PA, PC and PET) when they come first in the manuscript.
2) Line 45-47. ‘The range of polymers used in additive manufacturing includes thermoplastics (doi.org/10.5301/jabfm.5000343), thermosets (10.1021/acs.biomac.9b00941) , elastomers (doi.org/10.1038/s41427-019-0109-y), hydrogels (doi.org/10.3389/fmats.2020.00076), functional polymers, polymer blends (doi.org/10.1039/C7TC02534C), composites (10.1016/j.cej.2021.12854) and biological systems (10.1016/j.jsamd.2016.04.001), which already enable the creation of value-added materials.’ Please refers these recent work for the elements used in AM technology.
3) Introduction is very detailed but a little bit redundant. The authors can give more emphasis on the advantages of printing biodegradable biopolymer candidates (PLA and PHB) through FDM technology in this manuscript, instead of introducing the FDM technology for a wide range of polymers.
4) Why did the color change more significantly in experimental cohorts? Any relationship with degradation or thermal stability?
5) Please offer DSC curve of the tested samples
6) Multiple previous works (10.1016/j.actbio.2016.08.023; 10.1016/j.actbio.2018.03.011) have found that the mechanical properties of the biodegradable polymer can be impaired under hydrated condition. Since the materials can be recycled in this manuscript, will other environmental histories, particularly high humidity or buffer condition, will negatively affect the final mechanical properties of the blended biodegradable polymer? Please refer to these works or supplement new results to discuss the concern.
Author Response
Dear Reviewer, Thank you very much for your review. I have taken all your comments into account when correcting this article:
- added citations in the introduction
- introduction has been shorted
- added thermograms
- discussed effect of moisture
- regarding the color differences between experimental and industrial sample types, we believe that they are due to the different degree of degradation of the tested material during the blending process of the polymer blend on the experimental vs. industrial twin-screw extruder.
Reviewer 2 Report
Dear Authors,
The authors report the effect of multiple processing of biodegradable polymer blends based on PLA and PHB,
under the trade name NONOILEN, which were processed in laboratory as well as industrial conditions, to find
out about the final rheological and utility properties of these materials in 3D printing technology. From the results
they show in this work, they claimthat a biodegradable polymer blend based on PLA and PHB is a suitable
candidate for material recycling even in industrial processing conditions.
Although it’s an interesting work and there are many valuable points and references mentioned, however, there
are a couple of matters which need to be addressed before the manuscript can be accepted in “polymers” and
hence, I recommend the following comments.
1- The Manuscript needs thorough revision to improve the text quality and readability of work, some English
errors such as:
a- Page 2, Line 89: 3D printing is considered as an environmentally friendly processing...
b- Page 2, Line 100: ...to polymers that are either biobased or biodegradable or both.
2- The introduction section: At the moment, this section is too long with many of the sentences taken from
other references. The authors need to convince the readers why there is a need for this review article.
They may include statistics and references that prove an increase in research about their topic, at the
moment I’m not sure if I see this point in the introduction.
3- The Experimental section is incomplete: In this section, authors must include the materials and
reagents information, complete sample preparation, characterization methods and equipment used for
the performed characterizations (for e.g. SEM, DSC, etc.,).
** They all must be added in separate paragraphs to this section in the article.
** At the moment, this section is confusing and many information such as the following is either missing
or not well-written:
a- The raw materials are not really clear, it’s only mentioned, PLA, PHB and “other” additives (Page 6,
Line 270). What are “other” additives? There is no information about the reagent sources and their
quality / purity in the Experimental section.
b- The same issue with Polymer Blend preparation, it’s written “according to the recipe by Panara a.s.”
but no information on the recipe. (Page 6, Line 283)
c- No numbering and sub-numbering has been done on this section as well as other sections such as
1.1 or 1.2 etc... which makes this section confusing and unorganized.
4- Page 3, Line 128-131: The data are repeated and shall be re-written.
5- Page 4, Line 166: I believe it should be PLA / PHB and not PHA, according to the mentioned reference.
6- Page 6, Line 289-297: For the section “Multiple Extrusion of NONOILEN...”, would be much better if the
authors would provide a schematic scheme or a picture of the extruder and the procedure so that the
reader could follow easily.
7- The Figure sizes through the article are way too large, they must be adjusted as at the moment they are
taking too much space in the article.
8- Page 9, Line 401-402: The authors mention that in Figure 2, “”the largest decrease in complex viscosity
occurs during the preparation of the second filament – extrusion number 4 (I).”” whereas in the same
figure, it appears that the largest decrease in complex viscosity occurs for the first filament 2 (I). Why is
that?
9- Page 11, Line 454-456: It is mentioned that the lab samples show a decrease in PDI due to longer
retention times, whereas in the previous paragraph, authors discuss that the industrial samples
experience longer retention times. From what I understand, these two, contradict each other, why is that?
Is there any other reasoning except the retention time for low PDIs? As it appears this is the “reasoning”
for a lot of sections in this article.
10- Page 13, Line 474-476 & Page 16: Authors mention that samples degrade, and on page 16, they speak
about the glass transition temperature results. I wonder why the authors have not provided the original
thermal graphs (DSC and TGA) to the reader ? Would have been certainly helpful to see the results on
the graph as well.
11- Results & Discussion are written like a report with a minimum of text and a lot of figures. The conclusions
are not fully supported by the results, and some of them seem to be rather doubtful
Author Response
Dear reviewer, thank you for your review of this article. As part of the editing of the first manuscript, we tried to incorporate your comments:
- introduction has been shorted
- In the experimental part, we tried to better characterize the process of experimental and industrial processing, which we also proved by a schematic schema
- We have also included your comments Nos. 4 to 11 in the results section and in the conclusion
- It is not possible to state the exact composition of the tested blend because the blend is under patent protection
Reviewer 3 Report
Overall, this paper present a good finding that is worth for publication, however, some improvement is necessary before this paper is ready to be published.
Title: The title should be revise by using more specific term rather than “multiple processing” and “application properties” since this term is a general term.
Abstract:
- The abstract should consists of introduction, objective of study, methodology used, main findings (quantitative information is highly recommended), and conclusion. Please make sure all this item present in the abstract.
- The methodology is not clear, please revise by providing more accurate information regarding the preparation and testing conducted.
- The results is not clear, please provide accurate summary on the main finding based on the testing conducted.
Introduction:
- The introduction is too lengthy for an original research article. Generally, the author should only highlight the problem statement, introduction to the materials used in this study, introduction on the related process i.e FDM, summary on the previous research carried out using the similar materials, summary on the previous research doing polymer blending, summary on the previous work of recycling 3D filament, and followed by the research gap which highlighted the importance of this research. Please do not mix the information regarding the author work in the middle of the introduction. Short summary of the current work should only be placed at the end of introduction. Please dedicate one paragraph for each point mentioned above.
Materials and methodology:
- Please reorganize the materials and methodology part since it is quite confusing. In general, author should only provide few general sub-section under materials and methodology. Firstly, describe the materials used in this study. Secondly, sample preparation (here you should clearly describe the difference between both method for industrial and laboratory recycling), then followed by individual sub-section for every test conducted on the samples. Please avoid using number 1. Or 2. suddenly in the middle of the manuscript.
- Please clearly indicate which process is repeated for 9 times. It is quite hard to understand the sentence by single reading. For example author can write, “The extrusion and granulation process were repeated for 9 times for the same samples”
- Please state the name and country for all the machine use in this study.
- Please check for grammar error.
Results and Discussion
- Please make sure that auhor use consistent term i.e industrial/commercial. Please check all Figure/Axis caption for this term.
- Please consider to use paragraph to aid the reader to better understand the main topic to be discuss in each paragraph. Writing in one long paragraph is not appropriate to present your discussion.
- In the methodology, the Table indicate “Summary processing time” however, in the Figure, the author use “Measurement time/processing time/extrusion time”, please be consistent to avoid confusion.
- Figure 4 and Figure 6; the x-axis is confusing, please confirm how the sample based on the processing time is converted into average Mr?
- Figure 8 is confusing since the description (3rd, 5th, 7th, 9th) and picture (1,2,3,4) is not the same.
- Figure 14-16, why there is no comparison between industrial/laboratory samples?
- Please remove the trend line from all Figure, author can use connecting point with colour to enhance the clarity of the Figure.
- Overall the discussion is lacking of cross reference to support the argument made by author. Please improve on this by explaining the finding by supporting the argument with cross reference.
Conclusion
- Please provide accurate conclusion based on the main finding to be highlighted from the testing conducted on the samples. For example, the difference between industrial/experimental or rhte effect of extrusion time on the mechanical properties.
- The author write “There is also no negative impact on the strength characteristics of multiple processed samples” however the results show different finding. For example, it can be noted from Figure 11 that the yield strength were reduced for the experimental samples after 600 second.
- Please provide more accurate information in the conclusion.
Please send this manuscript for professional proofreading prior to resubmission.
Author Response
Dear reviewer, thank you for your review of this article. As part of the editing of the first manuscript, we tried to incorporate your comments:
- The title has been revised
- Introductoin has been shortened
- Materials and methodology have been reorganized and a diagram showing the differences between industrial and experimental recycling has been added for better understanding
- According to your comments, the result section of the article has also been redrafted and the conclusion has been modified in the same way
Round 2
Reviewer 2 Report
Except with the whole numbering system of the manuscript, about which, am still not very satisfied, the rest of the article is ok.